# A Snapshot Multi-Spectral Demosaicing Method for Multi-Spectral Filter Array Images Based on Channel Attention Network

**DOI:** 10.3390/s24030943

**Published:** 2024-02-01

**Authors:** Xuejun Zhang, Yidan Dai, Geng Zhang, Xuemin Zhang, Bingliang Hu

**Affiliations:** 1Xi’an Institute of Optics and Precision Mechanics, Chinese Academy of Sciences, Xi’an 710119, China; zhangxuejun2020@opt.ac.cn (X.Z.); daiyidan2020@opt.ac.cn (Y.D.); 2University of Chinese Academy of Sciences, Beijing 100049, China; 3Institute of Aerospace Science and Technology, School of Remote Sensing and Information Engineering, Wuhan University, Wuhan 430072, China; zhangxuemin@whu.edu.cn

**Keywords:** multi-spectral filter array, demosaicing, convolution neural network

## Abstract

Multi-spectral imaging technologies have made great progress in the past few decades. The development of snapshot cameras equipped with a specific multi-spectral filter array (MSFA) allow dynamic scenes to be captured on a miniaturized platform across multiple spectral bands, opening up extensive applications in quantitative and visualized analysis. However, a snapshot camera based on MSFA captures a single band per pixel; thus, the other spectral band components of pixels are all missed. The raw images, which are captured by snapshot multi-spectral imaging systems, require a reconstruction procedure called demosaicing to estimate a fully defined multi-spectral image (MSI). With increasing spectral bands, the challenge of demosaicing becomes more difficult. Furthermore, the existing demosaicing methods will produce adverse artifacts and aliasing because of the adverse effects of spatial interpolation and the inadequacy of the number of layers in the network structure. In this paper, a novel multi-spectral demosaicing method based on a deep convolution neural network (CNN) is proposed for the reconstruction of full-resolution multi-spectral images from raw MSFA-based spectral mosaic images. The CNN is integrated with the channel attention mechanism to protect important channel features. We verify the merits of the proposed method using 5 × 5 raw mosaic images on synthetic as well as real-world data. The experimental results show that the proposed method outperforms the existing demosaicing methods in terms of spatial details and spectral fidelity.

## 1. Introduction

### 1.1. Background

Multi-spectral images, which contain rich spectral and spatial information, have been utilized in a wide range of applications in different fields such as food safety inspection [1], medical diagnosis [2], precision agriculture [3], and target tracking [4,5]. A series of multi-channel imaging systems utilizing image sensors have been invented to meet the demands for the capture of multi-spectral images (MSIs). These systems can be classified into the following three categories: (i) single-camera multi-shot systems [6], (ii) multi-camera one-shot systems [7], and (iii) single-camera one-shot systems [8,9,10,11,12,13]. Single-camera multi-shot systems rely on precisely controlled environments and complex equipment, while multi-camera single-shot systems require optical system calibration [14]. Furthermore, single-camera multi-shot systems capture spectral–spatial information at multiple spectral bands by sequentially switching a specific optical filter for each band [15]. Therefore, this technology is unsuitable for capturing dynamic scenes of moving targets because of switching among filters is time-consuming. To overcome these shortcoming, single-camera one-shot systems, which are equipped with special multi-spectral filter arrays (MSFAs), have emerged to acquire spectral–spatial information simultaneously from a single shot [16,17]. The principle is to reduce the acquisition time by adding a multi-spectral color filter array in front of the camera sensor. The structural design has the merit of compacting the sensor size and taking snapshots, which leads to low spatial resolution since each pixel of an MSFA only captures spectral information of a specific band. The recovering procedure, called demosaicing, as shown in Figure 1, is the key step to obtaining a full-resolution and high-quality multi-spectral image cube [18]. As the number of spectral bands and the sparsity of sampling increase, the estimation of missing pixels becomes more difficult, which makes the demosaicing procedure of MSFAs more challenging. For all this, a variety of MSFA patterns and corresponding demosaicing methods [19,20,21] have been proposed to improve demosaiced image quality. Figure 2 shows three different MSFA patterns with 6 bands, 5 bands, and 25 bands.

### 1.2. Related Works

Brauers and Aach [22] proposed a fast linear interpolation demosaicing method for a six-band MSFA, as shown in Figure 2a. Their demosaicing method initially employs bilinear interpolation for each band based on the color difference between channels, which was developed for the field of demosaicing for color filter arrays (CFAs). The method assumes that the adjacent bands exhibit similarity in the spatial structures, such as textures or edges. The similarity is referred to as inter-band correlation. Nevertheless, the assumption that there is a strong correlation across all channels cannot be satisfied.

Monno et al. [11] proposed a demosaicing method for the five-band MSFA pattern, as shown in Figure 2b. The filters of the MSFA are, respectively, R, Cy, G, Or, and B. The filter G is arranged most densely, allowing it to preserve spatial structures in the image more intricately than the others. Therefore, the proposed demosaicing method utilizes the G band to interpolate other bands based on inter-band correlations.

Wang et al. [23] extended a discrete wavelet transform (DWT)-based CFA demosaicing to the MSFA demosaicing. Their approach independently reconstructs the high-frequency and low-frequency components of the image. The reconstruction of the low-frequency component employs intra-band interpolation, whereas the high-frequency component’s reconstruction follows the “substitution rule” introduced by Driesen and Scheunders [24]. The substitution rule replaces the high-frequency components of each band in an MSI with the high-frequency components corresponding to the middle-spectral channel, assuming it to be the sharpest one. However, this method strongly relies on inter-channel correlations, and the substitution rule is applicable only to Bayer CFA but not fully extendable to MSFAs.

Miao et al. [25] proposed a novel demosaicing method that is binary tree and edge-sensing (BTES) based. In an MSFA, the sampling positions of each channel are mapped onto nodes in a binary tree. They interpolate the unknown pixel iteratively by edge-aware interpolation to improve the quality of the resulting images. However, due to the fact that the estimation of missing pixels in BTES only utilize intra-channel correlation, the reconstructed image failed to retain spatial details.

Mihoubi et al. [26] involved the use of a pseudo-panchromatic image (PPI), which is the average spectral value of all spectral bands, to aid in the demosaicing process. Their approach assumes a strong correlation between the PPI and all channels. The proposed method combines the pseudo-panchromatic image with the original multi-spectral image to improve the accuracy and quality of the demosaicing process, resulting in more accurate multi-spectral images. Nevertheless, the assumption regarding the correlation between PPI and all channels in this method seems to be relatively weak, and it requires complex computations.

In recent years, convolutional neural networks (CNNs) have been employed as a data segmentation method for many low-level image processing problems, such as image deblurring [27], super-resolution [28,29,30], mechanical fault diagnosis [31,32], and denoising [33,34,35]. In recent years, many multi-spectral reconstruction methods have also been proposed, such as MST++ [36], HSCNN+ [37], and AWAN [38]. MST++, which achieves higher-performance metrics with fewer computational and parameter resources, employs a multi-stage spectral-wise transformer to perform spectral reconstruction. HSCNN+ is a CNN-based method for RGB image hyperspectral restoration. This method achieves more accurate solutions by deepening the network structure. AWAN is a deep self-adaptive weighted attention network that can obtain more precise MSI and better reconstruction quality. Similarly, all kinds of deep learning-based methods have been proposed for MSFA deosaicing. Shinda [39] proposed a deep deosaicing network that combines the deep residual network ResNet with three-dimensional (3D) convolution, the final results of which are superior to the pseudo-panchromatic image difference (PPID) method. However, the images generated by this approach may exhibit false color artifacts in areas of high contrast and brightness. The method proposed by Wisotzky [40] involves using raw sparse multi-spectral images (MSIs) cube as the input of the deep convolutional network (DcNet), maintaining the complete spatial information of the original image. However, performing standard convolution on sparse inputs can lead to some artifacts, making the convergence of the network more difficult. Many deep-learning-based methods often validate their effectiveness on publicly available datasets, but they rarely validate their performance on real-world captured images. Furthermore, improving the quality of real-world captured images using models trained on public datasets remains a challenging problem. In this paper, we assume that public datasets can be used to simulate images captured by a real camera based on camera parameters. The aim is to enhance the quality of real captured images by training better models.

### 1.3. Our Contribution

Multi-spectral imaging systems, which are based on MSFA, often capture objects of interest in complex environments, resulting in poor image quality. Therefore, the proposed demosaicing method must have strong robustness. Moreover, the above demosaicing methods have certain limitations in the quality of image restoration.

In this paper, based on the spectral and spatial correlations existing in mosaic images captured by the snapshot MSFA imaging sensors, we propose a deep neural network that combines channel attention mechanisms with CNNs, which is able to extract features automatically and protect important channel features, for multi-spectral images based on 25-band MSFA, as shown in Figure 2c. The proposed demosaicing method makes the following main contributions:We analyze the acquisition process of multi-spectral images, laying the theoretical groundwork for synthesizing the mosaic images and the radiance label images using the spectral sensitive functions (SSFs) of the IMEC camera we purchased and the available illuminants.We present a simple and feasible end-to-end deep convolution neural network that introduces the channel attention mechanism, which is able to adaptively adjust channel feature response and protect important channel features.The methodology we propose exhibits superior demosaicing performance on both simulated datasets and real-world scenarios compared to other existing methods, offering significant potential for the application of IMEC’s camera in both commercial and industrial sectors.

## 2. Observation Model

In this section, we perform the description of the observation model for the snapshot imaging systems based on MSFA. A single-sensor multi-spectral camera fitted with MSFA of *K* band provides a raw image $I^{MSFA}$ with M×N pixels. As depicted in Figure 3, at each pixel of a raw mosaic image $I^{MSFA}$, only one out of the *K* bands is available while the levels of the remaining bands are missing. Furthermore, we consider that a fully defined MSI with *K* bands ${\left\{G_{\lambda }\right\}}_{1}^{K}$ can be modulated through a series of sampling matrices ${\left\{X_{\lambda }\right\}}_{1}^{K}$. Finally, the raw mosaic image can be formulated as
(1)$$I^{MSFA}=\underset{\lambda =1}{\overset{K}{\Sigma }}X_{\lambda }\cdot G_{\lambda }$$
where · represents the pixel-wise product.

Our expectation is to use the CNN network to learn a mapping model *T* that can estimate a high-precision MSI. The entire process can be formulated as
(2)$$\overset{\wedge }{\theta }=arg\;min\;l^{MSI}\left(T\left(I^{MSFA};\theta \right),G\right)$$
where $l^{MSI}\left(\cdot \right)$ refers to the loss function, and θ represent the parameters of the networks.

## 3. Reference Image Simulation

We refer to the multi-spectral image $I={\left\{I^{j}\right\}}_{j=1}^{K}$, which is composed of K fully defined channels corresponding to *K* bands, as the reference image. The reference image is often utilized as a reference for evaluating the demosaicing quality, although it cannot be provided by multi-spectral cameras fitted with a single-sensor. To obtain the reference image, we employ the multi-spectral image formation model described in Section 3.1 to simulate the acquisition process. Subsequently, we will utilize the acquired reference image to generate spectral mosaic images, which serve as inputs to our network.

### 3.1. Multi-Spectral Image Formation

Assuming ideal optics and uniform spectral sensitivity of the sensors, the image for the *j*-th channel can be expressed as
(3)$$I^{j}=Q\left(\int_{\Omega }E\left(\lambda \right)\cdot R\left(\lambda \right)\cdot T^{j}\left(\lambda \right)d\lambda \right)$$
where the term E(λ) refers to the relative spectral power distribution of the light source, which uniformly illuminates all surface elements within the scene. Ω stands for the working spectral range. R(λ) represents the reflectance function of all surface elements. The camera captures the radiance spectrum represented by E(λ)·R(λ). This spectrum is then filtered based on the transmittance $T^{j}\left(\lambda \right)$ of the band *j*, which is centered at the wavelength $\lambda^{j}$. The pixel values in the image of the last *j*-th channel are determined by quantifying the received energy using the function *Q*.

### 3.2. Simulation of Radiance Data

In order to simulate the radiance data, we need (i) illumination data and (ii) reflectance. (i) We specifically consider three standard light sources (F12, A, and D65) to generate radiance data. Their relative spectral power distributions E(λ) are defined for all [420 nm, 1000 nm]. (ii) The TT-59 database consists of high-quality hyperspectral images, which are saved in the form of spectral reflectance. The reflectance is defined on 59 bands, ranging from 420 nm to 1000 nm at 10 nm intervals and centered at {420 nm, 430 nm, … 1000 nm}. Assuming linear continuity of reflectance, we can use linear interpolation of the reflectance data from the TT-59 database to get R(λ) for all integer wavelengths λ within the range of [420 nm, 1000 nm].

### 3.3. Multi-Spectral Image Simulation

In order to simulate the reference channels provided by the IMEC camera we purchased according to Equation (Equation 1), we also need transmittance $T_{IMEC}^{i}\left(\lambda \right)$ of the the IMEC camera. Transmittance $T^{i}\left(\lambda \right)$ differs based on the specific camera. The IMEC camera samples 25 bands with known transmittance $T_{IMEC}^{i}\left(\lambda \right)$. By summing discretely with d(λ)=1, Equation (Equation 1) transforms to
(4)$$I^{j}=Q\left(\underset{\lambda =420}{\overset{1000}{\Sigma }}E\left(\lambda \right)\cdot R\left(\lambda \right)\cdot T_{IMEC}^{i}\left(\lambda \right)\right)$$

## 4. Proposed Demosaicing Method

### 4.1. Network Framework

Multi-spectral demosaicing can be seen as an interpolation method that utilizes known pixel information from the raw mosaic image to estimate the missing pixel information. The bilinear interpolation, which is fast and easy to achieve, can effectively recover low-frequency information in the image. Furthermore, the CNN has the ability to reconstruct the high-frequency information of the image by learning the mapping relationship between the original mosaic image and the multi-spectral image. Therefore, we utilize the strengths of both to achieve the reconstruction process.

First of all, we regard the multi-spectral demosaicing process as an end-to-end learning task. The input to our network is the two-dimensional raw mosaic image generated by Section 3.3. Based on the spatial and spectral correlation existing in multi-spectral filter array mosaic images, we propose an end-to-end deep convolution neural network that introduces the channel attention mechanism to reconstruct an fully defined and high-quality MSIs. The proposed method first utilizes *K* sampling matrices to separate the band of the initial mosaic input image, as shown in Figure 4, generating *K* sparse single-spectral images. Sparse single-spectral images exhibit values exclusively at particular spectral band pixels, with information absent at the remaining pixels. The proposed method utilizes a 9×9 convolution filter, as shown in Equation (Equation 6), to perform a rough and quick reconstruction. The 9×9 convolution is performed independently on sparse spectral images for each spectral band. The low-frequency information of the images can be effectively recovered, but the details and textures of the images suffer from more severe loss. We utilize parallel convolution and a deep residual network (ResNet) to reduce artifacts in the initial demosaiced image. The efficient channel attention (ECA) module [41] is introduced into both the parallel convolution and ResNet to adaptively adjust channel feature response and protect important channel features. The ECA module eliminates the fully connected layer typically present in traditional attention mechanism modules. Instead, it directly employs a 1D convolution on the features following global average pooling for learning. Because experimental evidence has shown that convolution has excellent cross-channel information-gathering capabilities. The ECA module achieves the fusion of global contextual information by squeezing the feature maps. This step is accomplished by performing global average pooling, transforming the feature maps from a size of (N, C, H, W) to (N, C, 1, 1). Then, ECA calculates the size of the adaptive convolutional kernel using Equation (Equation 5) and maps the weights between 0 and 1 using the sigmoid activation function. In Equation (Equation 5), C represents the number of input channels, we set γ and b to 2 and 1. Finally, ECA performs element-wise multiplication between the reshaped weight values and the original feature maps to obtain feature maps with different weights. The diagram of the ECA module is shown in Figure 5. The detailed structure of the proposed method network is illustrated in Figure 6.
(5)$$k=\left|\frac{log_{2}\left(C\right)}{\gamma }+\frac{b}{\gamma }\right|$$
(6)$$F=1/25\left[\begin{array}{ccccccccc} 1 & 2 & 3 & 4 & 5 & 4 & 3 & 2 & 1\\ 2 & 4 & 6 & 8 & 10 & 8 & 6 & 4 & 2\\ 3 & 6 & 9 & 12 & 15 & 12 & 9 & 6 & 3\\ 4 & 8 & 12 & 16 & 20 & 16 & 12 & 8 & 4\\ 5 & 10 & 15 & 20 & 25 & 20 & 15 & 10 & 5\\ 4 & 8 & 12 & 16 & 20 & 16 & 12 & 8 & 4\\ 3 & 6 & 9 & 12 & 15 & 12 & 9 & 6 & 3\\ 2 & 4 & 6 & 8 & 10 & 8 & 6 & 4 & 2\\ 1 & 2 & 3 & 4 & 5 & 4 & 3 & 2 & 1 \end{array}\right]$$

The proposed network consists of three steps: initial demosaicing, convolutional network, and residual network. The initial spectral mosaic image is the input of the network. The process of the initial demosaicing is used to transform the input from 480×480×1 to 480×480×K. The value of *K* is set to 25, and it is determined by the number of spectral bands of the specific snapshot camera. The relationship between the initial demosaicing image and the reference image is utilized to train the whole network model, which is able to restore the high-frequency information of the multi-spectral image. The convolutional network uses three convolution kernels of different sizes to implement parallel convolution operations on the initial demosaiced image, respectively, which is able to obtain the shallow features of the multi-spectral image. The three paths of the convolutional network adopt convolution kernels of 1×1, 3×3, and 5×5, respectively, and the number of kernels is 128. Each path has the same input, which is processed in the same way: convolution operation, rectified linear unit (ReLU), convolution operation, and ECA. The convolution operation can automatically extract important features from multi-spectral images without any manual supervision. The main function of ReLUs is to improve the expression ability of the CNN model by making the network sparse and reducing the possibility of gradient disappearing. The ECA module with channel attention mechanism is added at the end of each convolutional pathway, allowing for the adaptive adjustment of channel feature responses and preserving important information. The results of the three paths are summed as the input to the residual network. The residual network [42] can effectively extract deep features from multi-spectral images while preserving important information. The unique architecture can effectively solve the problem of gradient vanishing and explosion. Each residual block has the same structure with skip connections, which consists of two convolutional layers with the filter size of 3×3, ReLU activation functions, and ECA. The number of extracted feature maps is 128. There are 12 residual blocks in the residual network. The output of the residual network performs one convolutional operation to obtain the final fully defined multi-spectral image. A 5×5 convolution kernel is used for convolution, and the number of images obtained through convolution should be consistent with the number of spectral bands. The convolution layer has a stride with padding to ensure that the size of the input and output arrays remains the same. In the proposed network structure, all convolution kernel coefficients are determined by learning. The ReLU function in the proposed network is f=max(x,0), which has the ability to improve the speed of network training [43].

### 4.2. Loss Function

In order to enable the trained network model to more accurately reconstruct the fully defined MSI, we design a weighted combination loss function to minimize the signal reconstrucion errors of the demosaiced image. We define the real image as *I* and define the final reconstructed images as $\overset{\wedge }{I}$. The overall loss function can be formulated as
(7)$$L_{demosaic}=L_{mse}+\alpha \cdot L_{wavelet}$$

Typically, the MSE (mean squared error) is commonly used as a loss function in the field of image processing because it encourages estimated pixel values to be closer to the ground truth. The MSE loss function can be formulated as
(8)$$L_{mse}=\frac{1}{N}\cdot \underset{p}{\overset{N}{\Sigma }}\;{∥I_{p}^{k}-I_{p}^{\overset{\wedge }{k}}∥}_{2}^{2}$$
where *k* and *p* represent the channel index and the pixel index, and *N* is the number of pixels in the image.

Morever, we introduced an additional edge loss to enhance the shaperness and texture richness of the final estimated images. We transform the final estimated images and the reference images into wavelet domain, respectively, and calculate their MSE after transformation in the high-frequency sub-bands. The edge loss can be calculated as
(9)$$L_{wavelet}=\frac{1}{N_{w}}\cdot \underset{q}{\overset{N_{w}}{\Sigma }}\;{∥w_{q}^{I}-w_{q}^{\overset{\wedge }{I}}∥}_{2}^{2}$$
where $w_{q}^{I}$ and $w_{q}^{\overset{\wedge }{I}}$ represent the *q*th wavelet coefficients of the final estimated images and the reference images, respectively. $N_{w}$ is the number of high-frequency wavelet coefficients of the image decomposed by stationary wavelet transform. We use Haar filters in wavelet transformation and set the transformation level to 2.

## 5. Datasets and Training

We use the TT-59 database [44] to evaluate the effectiveness of the proposed demosaicing method. Multi-spectral images in the database are saved in the form of spectral reflectance, which is more convenient for us in synthesizing the radiance data. The TT-59 database has 16 scenes, which are divided into 11 scenarios for training the models and 5 scenarios for evaluating the performance of the model. We utilize the SSFs of the IMEC camera we purchased to synthesize the raw images with a 5×5 mosaic, illustrated in Figure 2c, where the spectral bands of the IMEC camera were centered at $\lambda_{i}$ = 699, 713, 726, 739, 752, 766, 780, 794, 807, 821, 834, 847, 859, 872, 884, 896, 908, 919, 930, 941, 952, 962, 972, 982, and 992 nm.

In the practical training of the proposed multi-spectral demosaicing approach, in order to artificially augment the number of training samples, we use two CIE standard light sources (D65 and F12) and rotate $90^{\circ }$, $180^{\circ }$, $270^{\circ }$ during the training of the model. During the training of the network, we randomly extract 32 mosaic images with a size of 480 × 480 as a batch input. The optimizer of the network is Adam, and the learning rate is initialized to 0.0001 for all of the layers. We implemented the network model using the TensorFlow neural network framework. The deep learning model was trained for 500 epochs. In this paper, we used Python 3.6 to train the models, and all the experiments were conducted on a laboratory server with the configuration of an NVIDIA GeForce RTX 3090 12 GB GPU.

## 6. Experimental Results with Simulated Data and Real-Word Data

To validate the effectiveness of the model, we compare the proposed demosaicing method with three existing traditional demosaicing methods including weighted bilinear interpolation (WB), binary tree-based edge-sensing (BTES), pseudo-panchromatic image difference (PPID), and two deep-learning-based methods, including ResNet-3D [39] and deep convolutional network (DcNet) [40]. Furthermore, three metrics were utilized to represent the quality of the reconstructed multi-spectral image [45,46], which are the peak signal-to-noise ratio (PSNR), the structural similarity (SSIM), and the spectral angle mapper (SAM), respectively. The PSNR is to calculate the difference in pixel values between the reconstructed image and the reference image. The larger the value of PSNR between the two images, the better the quality of the reconstructed image. The SSIM is an image quality evaluation index that is utilized to measure the image degradation between the reconstructed image and the reference image. The value range of SSIM is [0,1]. The larger the SSIM value, the smaller the distortion degree of the reconstructed image. The SAM, which is widely used in pixel classification in multi-spectral imaging, is utilized to measures the spectral fidelity between the reconstructed image and the reference image by calculating the spectral angle. The smaller the value of SAM, the higher the spectral similarity between the reconstructed multi-spectral image and the reference image. We select two scenes Cloth3 and Spray in the database to visually compare the reconstructed images, which were generated by the above demosaicing methods. We crop and zoom in an area of 140×140 pixels from the reconstructed images for display. The visual comparison of reconstructed results by the above demosaicing methods is shown in Figure 7 and Figure 8.

As shown in Figure 7 and Figure 8, the edge of the image reconstructed by WB and BTES is highly blurred, and high-frequency information is seriously lost. The preservation of details in the PPID is significantly superior to WB and BTES, but the PPID still suffers from poor artifact prevention. The results from the ResNet-3D and DcNet are significantly superior to conventional methods, but still exhibit some artifacts. The image reconstructed of the proposed method provides better edge information with almost no artifacts and noise.

The PSNR and SSIM quantitative comparison results are shown in Table 1. The best results are highlighted in bold. Furthermore, The PSNR and SSIM values are calculated from the average of all test scenes in the database. Compared to other demosaicing methods in the spatial domain, the proposed demosaicing method has a significant improvement in both the PSNR and the SSIM. Table 1 indicates that the proposed demosaiking method outperforms the existing methods. The SAM comparison results are given in Table 2. In order to verify the effect of the proposed method, we randomly select four points from the reconstructed image to calculate SAM. The average SAM is reduced to 0.4168. Table 2 indicates that the proposed demosaiking method has an obvious superiority over other existing methods in spectral domain.

Due to the fact that the visual quality of an image cannot be strictly reflected by the PSNR/SSIM metrics, we compare the subjective quality of the demosaicing results on three test scenes from the *TT*-59 database using the error maps, which are the absolute errors between the demosaicing results and the ground truth. As shown in Figure 9, our method has superior demosaicing results in MSI images, which indicates that our method outperforms other demosaicing methods in terms of spatial accuracy. To further compare the superiority of the proposed method compared to other demosaicing methods in terms of spectral performance, we compared the absolute errors of all methods along the spectrum using the three scenes in Figure 9. As shown in Figure 10, the proposed method is much closer to the ground truth, which demonstrates higher spectral fidelity than other methods.

To evaluate the generalization ability of the proposed model, we select scene peppers in the CAVE database to visually compare the reconstructed images. We crop an area of 140×140 pixels from the reconstructed images for display. The reconstructed results by the above demosaicing methods are shown in Figure 11. As shown in Figure 11, The image reconstructed of the proposed method provides better edge information with almost no artifacts and noise. To further validate the effectiveness of our method, we conducted quantitative comparisons using three metrics: PSNR, SSIM, and SAM. The results are shown in Table 3 and Table 4. The quantitative comparison results indicate that the proposed method outperforms the other methods.

The proposed demosaicing method is applied in real-world images acquired by a real 5×5 snapshot camera. This camera has a spatial resolution of 409×217 (per band), and the frame rate can reach up to 170 data cubes per second (full sensor frame). As shown in Figure 12 and Figure 13, the proposed method is better than the traditional methods of WB and PPID. In terms of the details of the overall image, our method is also superior to ResNet-3D and DcNet.

## 7. Discussion

In this paper, we propose a snapshot multi-spectral demosaicing method for MSFA images based on the channel attention network. Extensive experiments indicate that the proposed demosaicing method has an obvious superiority over other existing methods in image quality and spectral domain. We also demonstrate that our method can guarantee real-time demosaicing performance in real-world shooting scenarios in ablation studies.

However, due to the limited number of training samples and the influence of the network architecture, the results of the proposed demosaicing method on real-world data still have some artifacts. How to more effectively utilize public datasets to simulate images captured based on camera parameters is also a factor. This is also a direction on which we will focus our future research efforts.

## 8. Ablation Studies

To demonstrate the effect of the proposed ECA module, we use the plain network without any attention as our base model, and then we study the networks with different attention in Table 5:(1)The base model without any attention, labeled as NT-NA;(2)The base model with ECA added to the convolutional network, labeled as NT-CN;(3)The base model with ECA added to the residual network, labeled as NT-RN;(4)The base model with ECA, which is our full model, is labeled as NT-FM.

All of these networks are trained under the same default settings for fair comparison. We test and average all the test images in the TT-59 database. Table 5 indicates that the proposed method are all better in terms of spatial accuracy and spectral fidelity than the other attention modules.

To further validate the effectiveness of the ECA module, we selected scenes from the dataset for visual quality comparisons. As shown in Figure 14, the proposed method is better than the other attention modules.

The running time and computational cost of the demosaicing method are of great importance when implemented on a realistic multi-spectral imaging system. Therefore, we compared the running time and GFLOPs of several deep-learning-based methods in Table 4. All experiments are implemented using the same machine (Intel Xeon Platinum 8260 CPU 2.40GHz, and NVIDIA GeForce RTX 3090 12 G GPU) and implemented on all test images of the TT-59 database. As shown in Table 6, the proposed method can guarantee real-time demosaicing performance.

## 9. Conclusions

In this paper, we present an end-to-end network based on the channel attention network to demosaic spectral mosaic images acquired by an MSFA-based imager. To achieve high-accuracy demosaicing of MSFA images, the proposed method takes advantage of the channel attention mechanism to protect important channel features, which is able to avoid artifacts and aliasing during the process of demosaicing. Our experimental results demonstrated that the proposed method generates more accurate demosaicing results in terms of spatial accuracy and spectral fidelity compared to other existing methods. It also serves as a universal multi-spectral demosaicing approach that can be adapted to different MSFA patterns with different spectral resolutions. Finally, We validated the effectiveness of the proposed method on real-world mosaic images with a pattern size of 5×5, which offers an alternative pathway for the real-time application of multi-spectral imaging in complex environments. Because the results of the proposed demosaicing method on real-world data still have some artifacts, we need to improve the network structure (e.g., change from bilinear interpolation to PPID in the initial demosaicing) in the future. In summary, the proposed method could be potentially utilized to build high-quality MSFA-based image acquisition systems working well in object detection, medical diagnosis, and food safety inspection applications.

## Figures and Tables

**Figure 1 sensors-24-00943-f001:**
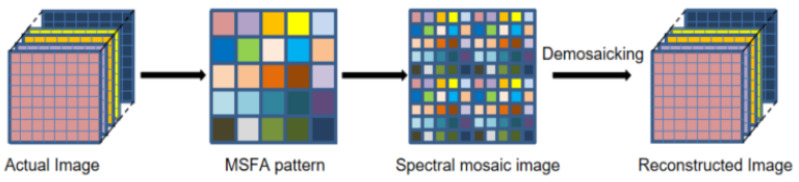
The procedure of multi-spectral demosaicing.

**Figure 2 sensors-24-00943-f002:**
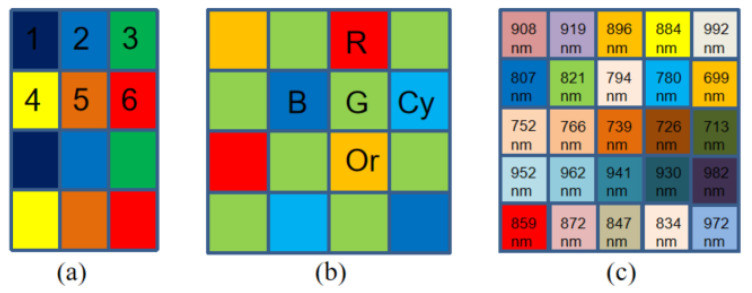
Several different MSFA patterns: (**a**) Brauers and Aach’s MSFA; (**b**) Monno et al.’s MSFA; (**c**) our MSFA.

**Figure 3 sensors-24-00943-f003:**
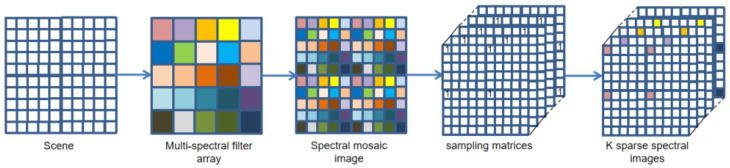
The model of a snapshot mosaic imaging system fitted with an MSFA.

**Figure 4 sensors-24-00943-f004:**
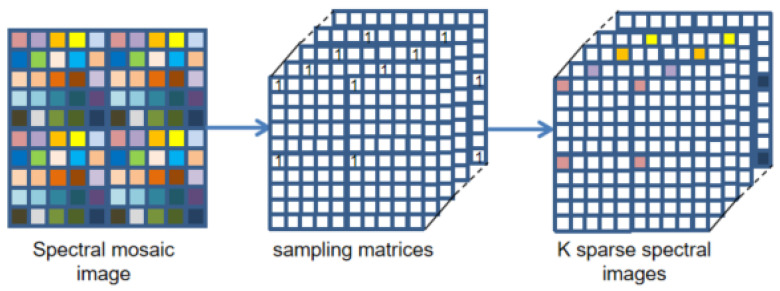
Band separation.

**Figure 5 sensors-24-00943-f005:**
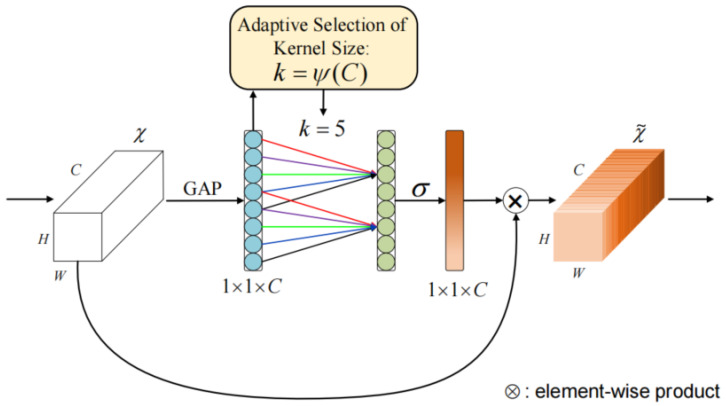
A diagram of the efficient channel attention (ECA) module.

**Figure 6 sensors-24-00943-f006:**
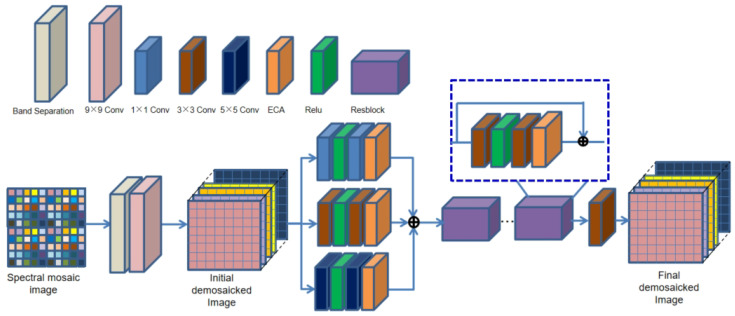
A schematic diagram of the proposed demosaicing method.

**Figure 7 sensors-24-00943-f007:**
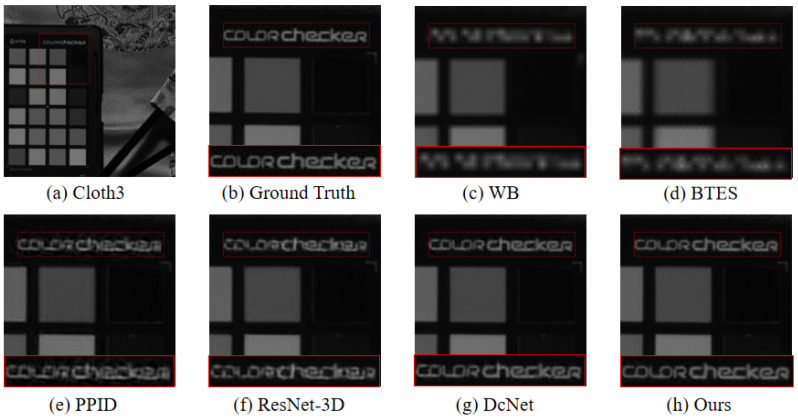
The visual quality comparison results of different demosaicing methods for cloth3 scenes. (**a**,**b**) is the reference image and the ground truth. (**c**) is the result of the WB. (**d**) is the result of the BTES. (**e**) is the result of the PPID. (**f**) is the result of the ResNet-3D. (**g**) is the result of the DcNet. (**h**) is the result of ours.

**Figure 8 sensors-24-00943-f008:**
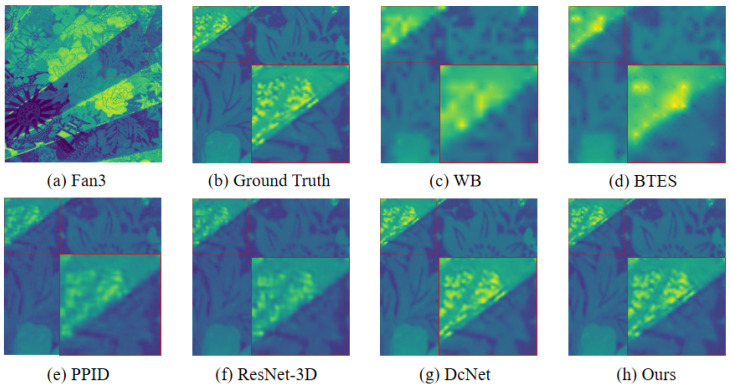
The visual quality comparison results of different demosaicing methods. (Images are converted to sRGB for display). (**a**,**b**) is the reference image and the Ground Truth. (**c**) is the result of the WB. (**d**) is the result of the BTES. (**e**) is the result of the PPID. (**f**) is the result of the ResNet-3D. (**g**) is the result of the DcNet. (**h**) is the result of ours.

**Figure 9 sensors-24-00943-f009:**
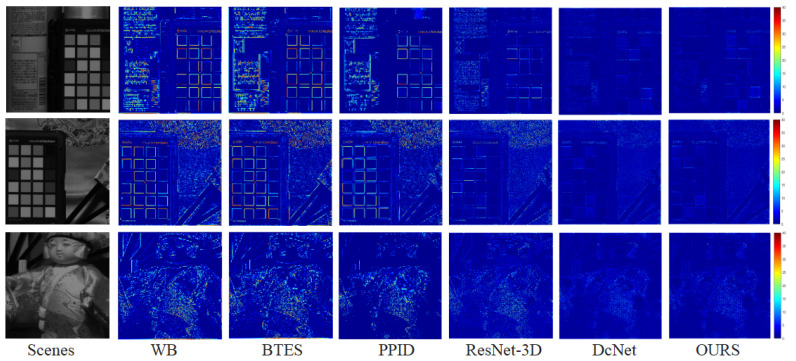
Visual quality comparison of representative scenes of the *TT*-59 database at 699 nm. The error maps for WB/BTES/PPID/ResNet-3D/DcNet/our demosaicing results.

**Figure 10 sensors-24-00943-f010:**
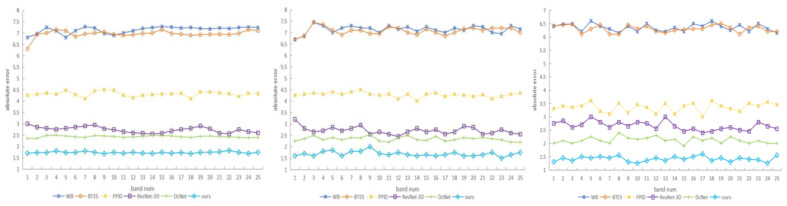
The absolute error between the demosaicing results and the ground truth of the scenes in Figure 9 along the spectrum for all methods.

**Figure 11 sensors-24-00943-f011:**
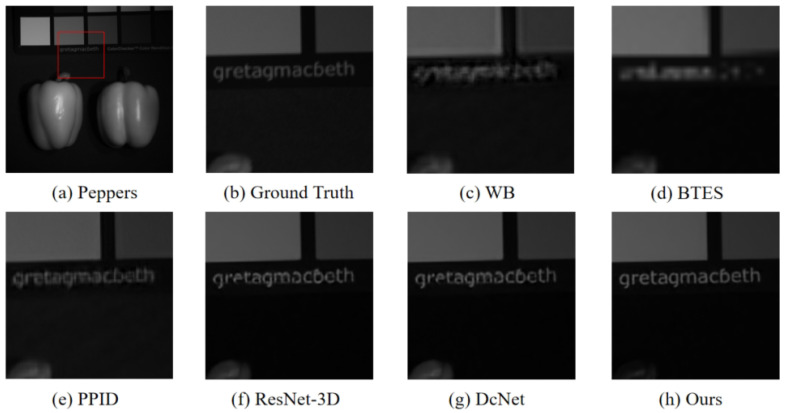
The visual quality comparison results of different demosaicing methods for peppers scenes. (**a**,**b**) is the reference image and the Ground Truth. (**c**) is the result of the WB. (**d**) is the result of the BTES. (**e**) is the result of the PPID. (**f**) is the result of the ResNet-3D. (**g**) is the result of the DcNet. (**h**) is the result of ours.

**Figure 12 sensors-24-00943-f012:**
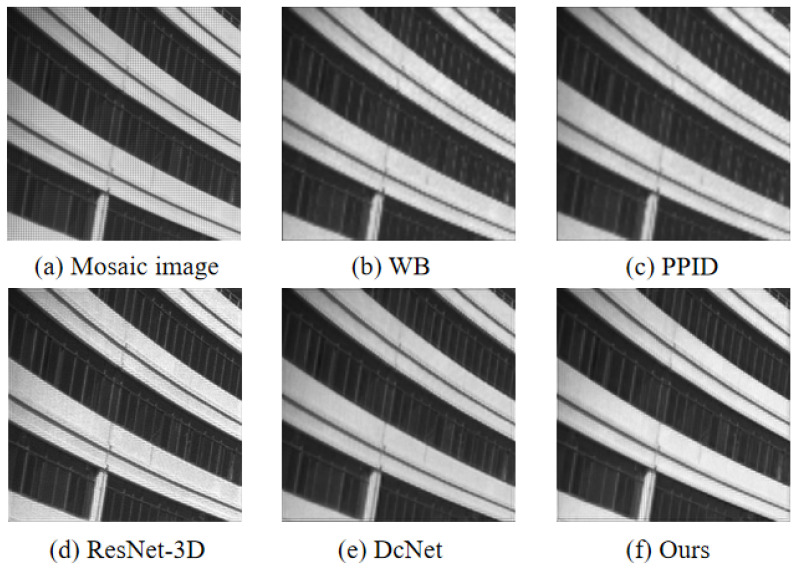
The visual results of a building using different demosaicing methods at 699 nm. (**a**) is the mosaic image. (**b**) is the result of the WB. (**c**) is the result of the PPID. (**d**) is the result of the ResNet-3D. (**e**) is the result of the DcNet. (**f**) is the result of ours.

**Figure 13 sensors-24-00943-f013:**
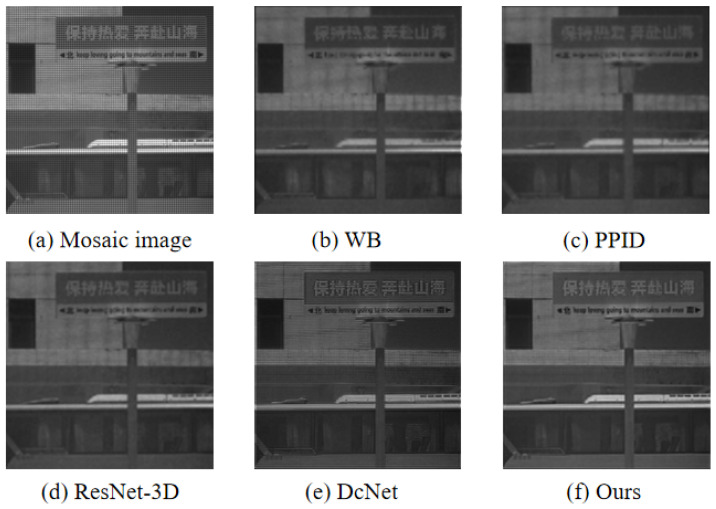
The visual results of a signboard using different demosaicing methods at 699 nm. (**a**) is the mosaic image. (**b**) is the result of the WB. (**c**) is the result of the PPID. (**d**) is the result of the ResNet-3D. (**e**) is the result of the DcNet. (**f**) is the result of ours.

**Figure 14 sensors-24-00943-f014:**
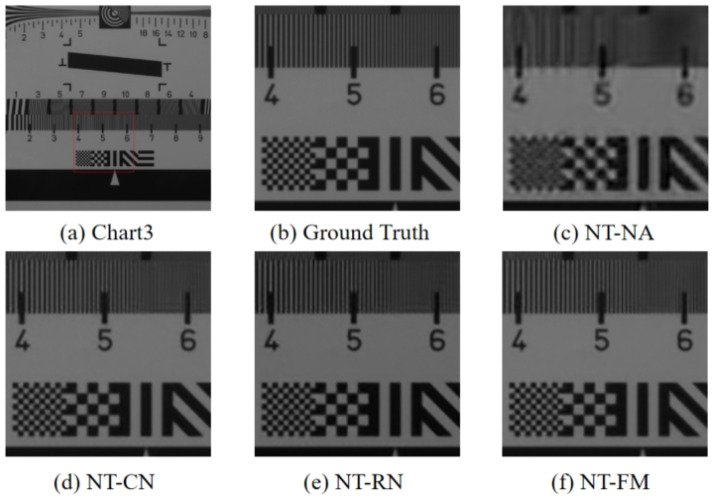
The visual results of different cases.

**Table 1 sensors-24-00943-t001:** Demosaicing results (PSNR/SSIM) for three typical scenarios and the average results for all test scenarios on the TT-59 dataset. (Bold indicates the best result).

Methods	Spray		Cloth3		Doll2		Average of All Test	
	PSNR ↑	SSIM ↑	PSNR ↑	SSIM ↑	PSNR ↑	SSIM ↑	PSNR ↑	SSIM ↑
WB	26.60	0.9006	25.42	0.8998	26.88	0.9068	26.42	0.9016
BTES	26.72	0.9076	25.78	0.9014	27.56	0.9106	27.64	0.9048
PPID	35.46	0.9810	33.78	0.9624	36.78	0.9764	35.78	0.9784
ResNet-3D	41.87	0.9992	39.41	0.9975	40.26	0.9970	41.53	0.9978
DcNet	42.76	0.9993	41.12	0.9980	41.77	0.9972	41.98	0.9982
OURS	**45.87**	**0.9999**	**42.50**	**0.9988**	**43.55**	**0.9984**	**44.88**	**0.9990**

**Table 2 sensors-24-00943-t002:** Demosaicing results (SAM) at different pixel points for doll2 scenes. (Bold indicates the best result).

Image	Methods	Point 1	Point 2	Point 3	Point 4	Average
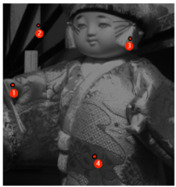	WB	1.2973	1.1768	1.1618	1.6128	1.2482
	BTES	1.1236	1.1752	1.1527	1.5023	1.2378
	PPID	0.8952	0.8681	0.8486	0.8749	0.8254
	ResNet-3D	0.8011	0.8340	0.7981	0.8292	0.8022
	DcNet	0.6420	0.6690	0.7447	0.6791	0.6814
	OURS	**0.4266**	**0.4331**	**0.4720**	**0.4154**	**0.4168**

**Table 3 sensors-24-00943-t003:** Demosaicing results (PSNR/SSIM) for three typical scenarios and the average results for all test scenarios on the cave dataset. (Bold indicates the best result).

Methods	Sponges		Paints		Feathers		Average of All Test	
	PSNR ↑	SSIM ↑	PSNR ↑	SSIM ↑	PSNR ↑	SSIM ↑	PSNR ↑	SSIM ↑
WB	25.58	0.9112	26.18	0.9018	26.46	0.9132	25.84	0.9106
BTES	25.88	0.9178	26.72	0.9124	26.88	0.9186	26.95	0.9162
PPID	36.62	0.9788	34.22	0.9728	37.46	0.9652	36.12	0.9720
ResNet-3D	40.26	0.9986	39.58	0.9964	39.86	0.9972	40.84	0.9980
DcNet	41.52	0.9990	40.46	0.9986	41.04	0.9982	41.08	0.9986
OURS	**44.12**	**0.9997**	**41.86**	**0.9986**	**43.20**	**0.9988**	**43.24**	**0.9989**

**Table 4 sensors-24-00943-t004:** Demosaicing results (SAM) at different pixel points for peppers scenes. (Bold indicates the best result).

Image	Methods	Point 1	Point 2	Point 3	Point 4	Average
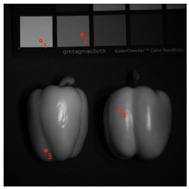	WB	1.3820	1.2526	1.2764	1.5208	1.3206
	BTES	1.2960	1.2542	1.2328	1.4856	1.2726
	PPID	0.9042	0.8528	0.8664	0.8812	0.8652
	ResNet-3D	0.8324	0.8226	0.8168	0.8456	0.8288
	DcNet	0.7812	0.7416	0.7852	0.7018	0.7182
	OURS	**0.5126**	**0.4864**	**0.4912**	**0.4328**	**0.4418**

**Table 5 sensors-24-00943-t005:** The ablation study of the network with different attention. (Bold indicates the best result).

Case	PSNR	SAM
NT-NA	41.56	0.5436
NT-CN	42.88	0.5324
NT-RN	42.86	0.5312
NT-FM	**44.88**	**0.5132**

**Table 6 sensors-24-00943-t006:** The demosaicing performance comparison of different methods.

Methods	Running Times (ms)	GFLOPs
ResNet-3D	2.87	932.4
DcNet	2.16	50.6
Ours	2.46	68.2

## Data Availability

The public dataset used in this paper can be downloaded at: http://www.ok.sc.e.titech.ac.jp/res/MSI/MSIdata59.html (accessed on 1 October 2023).

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
