# Peer review of "A Snapshot Multi-Spectral Demosaicing Method for Multi-Spectral Filter Array Images Based on Channel Attention Network"

_sensors, 2024, doi:10.3390/s24030943_

Round 1

Reviewer 1 Report

Comments and Suggestions for Authors

This paper This paper proposes a snapshot multi-spectral demosaicing method for MSFA images based on Channel attention Network.

The topic is interesting and the method is novel.

I have the following comments:

1) The motivation of this study should be improved. In abstract, the author declare that the existing demosaicing methods will produce adverse artifacts and aliasing. More explanation should be given regarding the declaration.

2) Why did the authors choose the network framework in Section 4?

3) What about the computational efficiency of the developed approach?

4) More relaetd approaches such as  "Cross-modal Fusion Convolutional Neural Networks with Online Soft Label Training Strategy for Mechanical Fault Diagnosis"; "CFCNN: A novel convolutional fusion framework for collaborative fault identification of rotating machinery".

5) The mechanism of the Loss function should be described in more detail

6) Future work should be prospected in the conclusion section.

Reviewer 2 Report

Comments and Suggestions for Authors

The author has developed a multispectral imaging system based on MSFA and utilized deep learning methods to eliminate mosaics. The experimental results presented in the manuscript demonstrate that the author's proposed method is effective, as it can effectively remove mosaics and reconstruct multispectral images. However, there are still some issues with the manuscript that need to be addressed before it can be officially accepted, as follows:

  1. In the introduction section, while there is a review of current work on de-mosaicking, there needs to be more discussion of mainstream methods for multispectral reconstruction, particularly computational reconstruction techniques such as MST++, HSCNN+, AWAN, etc. It is recommended that the author include these classic reconstruction networks in the introduction review section.
  2. In the theoretical part, the author's introduction of the ECA module requires a more detailed analysis. It would be best to start from the level of mathematical principles and provide corresponding theoretical deductions to illustrate its necessity.
  3. In the experimental section, the manuscript does not specify the type of camera used for acquisition. It only provides a general description of the IMEC camera without disclosing specific resolution or frame rate information. It is recommended that the author provide these specific details.
  4. Detailed information about network training is insufficient, including the number of images used for training, image size for training, and convergence method used.
  5. There are also doubts regarding the dataset used. While the TT-59 dataset used in the manuscript is acceptable as it comes from a manuscript published in the IEEE Sensor Journal in 2019, public datasets are already available in this field, such as ARAD-HS, CAVE, etc. It is recommended that the author consider testing the proposed network on public datasets to verify better whether the proposed network performance is state-of-the-art (SOTA).
  6. The ablation experiment has too little data; it is recommended that the author provide corresponding reconstructed image data.

In summary, the manuscript of this manuscript has yet to reach the level of formal publication. It is recommended that the author carefully revise it to address these concerns.

Comments on the Quality of English Language

There is still room for improving English.

Reviewer 3 Report

Comments and Suggestions for Authors

Thanks

Reviewer 4 Report

Comments and Suggestions for Authors

As a reviewer and a reader, I only have some concern about the paper writing as listed below.

1.       Line 41: more challenge->more challenging

2.       Demosaicing (Line 46) or demosaicking (Line 42) need to be fixed so that they are consistent.

3.       Line 53: “The filters of the MSFA are respectively B, Cy, G, Or, and B.” Would they be R, Cy, G, Or, and B?

4.       Line 72: “Mihoubi et al. [23] involves the use of…” Why is a present tense used here while a past tense is used in previous corresponding paragraphs (Line 45, 52, 57, and 66)? Please make them consistent.

5.       Make sure your references are well introduced. For example, related work (Line 82): “super-resolution [26–28]” should be [25-27]?

6.       Figure 2 is not clear to readers, needing more detailed explanations.

7.       Line 116: “describtion”-> description

8.       Line 117-119: please rewrite it.

9.       Line 222 defined bar{I} and hat{I} for the initial image and the final image, respectively. However, in the following context, bar{I} didn’t appear. Should the “I” appearing in Equations (7) and (8) be bar{I}?

Comments on the Quality of English Language

1.       Line 41: more challenge->more challenging

2.       Demosaicing (Line 46) or demosaicking (Line 42) need to be fixed so that they are consistent.

3.       Line 53: “The filters of the MSFA are respectively B, Cy, G, Or, and B.” Would they be R, Cy, G, Or, and B?

4.       Line 72: “Mihoubi et al. [23] involves the use of…” Why is a present tense used here while a past tense is used in previous corresponding paragraphs (Line 45, 52, 57, and 66)? Please make them consistent.

7.       Line 116: “describtion”-> description

8.       Line 117-119: please rewrite it.

Round 2

Reviewer 1 Report

Comments and Suggestions for Authors

This paper is ok.

Author Response

Thank you very much for your review!  Your comments have been incredibly beneficial to me, and I will continue to strive for improvement in the future.

Reviewer 2 Report

Comments and Suggestions for Authors

The authors have diligently addressed the concerns raised by the reviewers, resulting in a revised manuscript that effectively responds to the questions and alleviates the reviewers' concerns. In summary, the current manuscript has substantially reached the standards for publication in your esteemed journal, and we concur with the recommendation for its formal acceptance.

Comments on the Quality of English Language

There is still room for improving English.

Author Response

(The authors gave the same response as above.)

Reviewer 3 Report

Comments and Suggestions for Authors

Thnaks

Author Response

(The authors gave the same response as above.)
